# Discovery and Genomic Function of a Novel Rice Dwarf-Associated Bunya-like Virus

**DOI:** 10.3390/v14061183

**Published:** 2022-05-29

**Authors:** Dan Wang, Shuai Fu, Hongyue Wu, Mengji Cao, Li Liu, Xueping Zhou, Jianxiang Wu

**Affiliations:** 1State Key Laboratory of Rice Biology, Institute of Biotechnology, Zhejiang University, Hangzhou 310058, China; 11716067@zju.edu.cn (D.W.); fushuai@zju.edu.cn (S.F.); hongyue.wu@immorna.com (H.W.); 2National Citrus Engineering and Technology Research Center, Citrus Research Institute, Southwest University, Chongqing 400712, China; caomengji@cric.cn; 3The Department of Applied Engineering, Zhejiang Economic and Trade Polytechnic, Hangzhou 310018, China; liul@zjiet.edu.cn; 4State Key Laboratory for Biology of Plant Diseases and Insect Pests, Institute of Plant Protection, Chinese Academy of Agricultural Sciences, Beijing 100193, China

**Keywords:** novel virus, bunyavirus, movement protein, RNA silencing suppressor

## Abstract

Bunyaviruses cause diseases in vertebrates, arthropods, and plants. Here, we used high-throughput RNA-seq to identify a bunya-like virus in rice plants showing the dwarfing symptom, which was tentatively named rice dwarf-associated bunya-like virus (RDaBV). The RDaBV genome consists of L, M, and S segments. The L segment has 6562 nt, and encodes an RdRp with a conserved Bunya_RdRp super family domain. The M segment has 1667 nt and encodes a nonstructural protein (NS). The complementary strand of the 1120 nt S segment encodes a nucleocapsid protein (N), while its viral strand encodes a small nonstructural protein (NSs). The amino acid (aa) sequence identities of RdRp, NS, and N between RDaBV and viruses from the family *Discoviridae* were the highest. Surprisingly, the RDaBV NSs protein did not match any viral proteins. Phylogenetic analysis based on RdRp indicated that RDaBV is evolutionarily close to viruses in the family *Discoviridae*. The PVX-expressed system indicated that RDaBV N and NS may be symptom determinants of RDaBV. Our movement complementation and callose staining experiment results confirmed that RDaBV NSs is a viral movement protein in plants, while an agro-infiltration experiment found that RDaBV NS is an RNA silencing suppressor. Thus, we determined that RDaBV is a novel rice-infecting bunya-like virus.

## 1. Introduction

Bunyaviruses are often transmitted by vectors and can infect vertebrates, arthropods, and/or plants [1]. In 2018, the International Committee on Taxonomy of Viruses (ICTV) reclassified bunyaviruses in the order *Bunyavirales* [2]. This order currently contains 14 families: *Arenaviridae*, *Cruliviridae*, *Fimoviridae*, *Hantaviridae*, *Leishbuviridae*, *Mypoviridae*, *Nairoviridae*, *Peribunyaviridae*, *Phasmaviridae*, *Phenuiviridae*, *Tospoviridae*, and *Wupedeviridae*, and two new families (*Discoviridae, Tulasviridae*) created by ICTV in July 2021 [3]. Virions of most bunyaviruses, except for non-enveloped tenuiviruses, are enveloped, spherical, and 80–120 nm in diameter [4]. Genomes of bunyaviruses are typically tripartite, consisting of large (L), medium (M), and small (S) segments [5]. The L segment encodes an RNA-dependent RNA polymerase (RdRp). The M segment encodes a glycoprotein (G) or/and a nonstructural protein (NS) [6,7]. The S segment of most bunyaviruses encodes a nucleocapsid protein (N), while the S segment of other bunyaviruses encodes N and other nonstructural proteins [6,7]. As the largest negative RNA virus order, it contains many newly identified bunyaviruses [8,9,10]. For example, Yu et al. reported a new bunyavirus from patients who presented with high fever, thrombocytopenia, leukocytopenia, and multiorgan dysfunction [8]. In 2016, Dong and colleagues identified a new bunyavirus in moribund oriental shrimp (*Penaeus chinensis*) on a farm in China [9]. Another novel bunyavirus was found in a *Euproctis pseudoconspersa* population obtained from the field [10].

Unlike other microorganisms, viruses cannot be analyzed using any universal conserved sequences, which makes it difficult to identify a novel virus [11]. In recent years, high-throughput sequencing (HTS), also known as next-generation sequencing (NGS) or deep sequencing, has proven to be an effective method of identifying new viruses [12]. The wide application of deep RNA-sequencing (RNA-seq) technology in new virus identification has facilitated the discovery of thousands of new viruses in recent years [13]. From 2017 to 2021, the total number of viruses recorded by ICTV increased from 3000 to more than 9000 [14,15]. In 2017, Yang et al. identified rice stripe mosaic virus on rice plants in China using RNA-seq [16]. Kraberger and colleagues reported rice latent viruses 1 and 2 on Australian wild rice plants in 2017 [17]. Our laboratory recently reported a rice curl dwarf-associated picornavirus in field-collected rice plants through RNA-seq [18].

Rice (*Oryza sativa*) is one of the most important food crops in the world and is planted mainly in Asian countries, including China [19,20]. Viral infections are serious threats to the rice industry [20]. As of 2021, a total of 17 rice viruses have been reported in the world [21]. Among them, rice dwarf virus (RDV), rice black-streaked dwarf virus (RBSDV), southern rice black-streaked dwarf virus (SRBSDV), rice gall dwarf virus (RGDV), rice stripe mosaic virus (RSMV), rice ragged stunt virus (RRSV), and rice stripe virus (RSV) have been found in paddy fields in China, seriously harming rice production in China [20,22]. Most of them can coinfect rice plants, leading to more serious symptoms [23]. For instance, SRBSDV can commonly co-infect with RRSV in the field [23]. This synergistic infection phenomenon increases the difficulty of controlling rice viruses. During the surveys of rice viral diseases in paddy fields in China, we found many rice plants showing virus-like symptoms but without any known rice virus. Therefore, we suspect that there is/are new rice virus(es) in these rice plants. Furthermore, identifying rice viruses in different rice cultivation regions is important for safe rice production.

In 2019, during the surveys of rice viral diseases in paddy fields in Zhejiang province, China, we found some rice plants with virus-like symptoms of dwarfing, reduced tillering, and blighted grain, and we collected these rice plants. RT-PCR detection results showed no known viruses in these field-collected rice plants. Through high-throughput RNA-seq analysis, we identified a new bunyavirus in these samples that was tentatively named rice dwarf-associated bunya-like virus (RDaBV). The RDaBV genome consists of L, M, and S segments. A phylogenetic analysis using RdRp amino acid sequences indicated that RdaBV is evolutionarily close to viruses from the family *Discoviridae*, which belongs to the order *Bunyavirales*. We also analyzed the functions of RDaBV-encoded proteins and found that the NSs protein has a movement function in plant cells and that the NS protein can act as an RNA silencing suppressor. Both RDaBV N and NS may be the symptom determinants of RDaBV. This study expands our knowledge of viral populations in rice fields in China and can help establish an effective strategy for controlling this disease in rice fields.

## 2. Materials and Methods

### 2.1. Rice Sample Collection, RNA-Seq, and De Novo Assembly

During the surveys of rice viral diseases in 2019, rice plants showing dwarfing, reduced tillering, and blighted grain were found and sampled from rice paddies in Zhejiang province, China. Total RNA was isolated from these rice samples using TRIzol reagent (Invitrogen, Carlsbad, United States). The extracted total RNA samples were used for cDNA library constructions, which were performed by the Zhejiang TianKe High-Technology Development Co., Ltd. (Hangzhou, China), followed by sequencing on an Illumina HiseqTM 4000 sequencing system (Illumina, San Diego, CA, USA), as previously described [18]. Low-quality sequencing reads were removed using the CLC Genomics Workbench 9.5 (Qiagen, Valencia, CA, USA); high-quality reads were used to map to the rice genome database (http://rice.plantbiology.msu.edu/, accessed on 13 October 2019). The none-plant specific reads were then *de novo* assembled using the Trinity v2.3.2 software [24].

### 2.2. Reverse Transcription-Polymerase Chain Reaction (RT-PCR) and 3′ and 5′ Rapid Amplification of the cDNA Ends (3′/5′RACE)

For viral genome amplification, total RNA was extracted from the collected rice samples and reverse-transcribed into cDNAs using the reverse transcriptase M-MLV (TaKaRa, Kusatsu, Japan) according to the manufacturer’s instructions. Super-fidelity PCR was performed using the KOD One PCR Master Mix (TOYOBO, Osaka, Japan) and specific primers (Appendix A). The rapid amplification of cDNA ends (RACE) was performed using the HiScript-TS 5/3′ RACE Kit (Vazyme, Nanjing, China), according to the manufacturer’s instructions, to amplify the 3′ and 5′ end sequences of the newly identified virus. The PCR products were individually sequenced and used to assemble the complete viral genome.

To survey viral infections in the paddy fields, total RNA from field rice plants was reverse-transcribed using the ReverTra Ace qPCR RT Master Mix with gDNA Remover (TOYOBO, Osaka, Japan) according to the manufacturer’s instructions. PCR reactions were performed to detect RDaBV using the Green Taq Mix (Vazyme, Nanjing, China) and the specific forward and reverse primer pair, RDaBV-Detection-F/R (Appendix A). Each PCR reaction system consisted of 10 µL of 2 × Green Taq Mix, 1 µL of each primer (10 µmol/L), 7 µL of sterile deionized water, and 1 µL of cDNA. PCR reaction conditions were 2 min at 94 °C; 30 cycles of amplification at 94 °C for 30 s, 54 °C for 30 s, and 72 °C for 30 s; the final extension step was performed at 72 °C for 5 min.

### 2.3. Analyses of Viral Genome and Proteins

Open reading frames (ORFs) in the viral genomic RNA of the new virus were predicted using the NCBI ORFfinder software (http://www.ncbi.nlm.nih.gov/orffinder, accessed on 18 December 2019) and using the sequence of the Plasmopara viticola lesion-associated mycobunyavirales-like virus 8 as a model [25]. Three viral genome segments of the new rice virus obtained in this study were used to perform BLASTx and BLASTp searches in the NCBI databases. In addition, the conserved domains in the viral proteins were identified using the Conserved Domain Search Service (CD-Search) software in the NCBI (https://www.ncbi.nlm.nih.gov/Structure/cdd/wrpsb.cgi, accessed on 10 January 2020) [26].

For homology analyses of genomic RNAs or predicted proteins of the new virus, we downloaded the nucleotide and amino acid sequences of representative species from all fourteen families of the order *Bunyavirales*. Detailed information on these sequences is provided in Appendix A. The sequence identities were aligned using the DNAMAN7.0 software (https://www.lynnon.com/dnaman.html, accessed on 10 January 2021).

### 2.4. Phylogenetic Analysis

Genome sequences of representative viruses in different families of the order *Bunyavirales* were downloaded from the GenBank database. Detailed information on these viruses is provided in Appendix A. Amino acid (aa) sequences of RdRps of these viruses, including RDaBV, were aligned using the ClustalW method in the MEGA X software. The phylogenetic analysis was performed using the maximum-likelihood method in the MEGA X software with 1000 bootstrap replicates [27]. The resulting phylogenetic tree was edited using the iTOL online tool (https://itol.embl.de/, accessed on 10 January 2021).

### 2.5. Plasmid Construction

*NS*, *N*, and *NSs* protein genes of the newly identified virus were individually amplified from the diseased rice plants with RT-PCR using the specific primers listed in Appendix A. The resulting genes were subcloned into the expression vector pCambia 1300-Flag, pCambia 1300-GFP, or pGR106 using the ClonExpress II One Step Cloning Kit (Vazyme, Nanjing, China) according to the manufacturer’s instructions.

### 2.6. Agroinfiltration Assay

To identify the RNA silencing suppressor of RDaBV, the constructed vectors were confirmed by nucleotide sequencing and then transformed into the *Agrobacterium tumefaciens* strain EHA105. Leaves of line 16c transgenic *Nicotiana benthamiana* were co-infiltrated with *A. tumefaciens* harboring the GFP gene and *A. tumefaciens* harboring the *N*, *NSs*, *NS*, tomato bushy stunt virus (TBSV) *P19* or *GUS* gene, respectively. GFP fluorescence in co-infiltrated leaves was observed and photographed under a UV lamp (UV Products, Upland, CA, USA).

To identify the symptom determinants, RDaBV *N*, *NSs*, and *NS* were separately subcloned into the PVX heterologous expression vector pGR106. The resulting recombinant expression vectors were transformed into the *A. tumefaciens* strain GV3101 using the freeze–thaw method as previously described [28]. *N. benthamiana* leaves were infiltrated with *A. tumefaciens* harboring the PVX-base expression vector. At 7 days post-agro-infiltration (dpi), the agro-infiltrated *N. benthamiana* and its leaves were observed and photographed.

### 2.7. Agrobacterium-Mediated Movement Complementation Experiment

An expression vector expressing a movement-defective potato virus X (i.e., PVX-GFPΔp25) was obtained from a previously published source [29]. The *A. tumefaciens* strain GV3101 harboring the vector PVX-GFPΔp25 was prepared and diluted to OD_600_ = 0.001 in an infiltration buffer (10 mM MgCl_2_, 10 mM MES (pH 5.8) and 100 μM acetosyringone), as previously described [30], and then mixed 1:1:1 (*v*/*v*/*v*) with *A. tumefaciens* cells harboring a vector expressing the TBSV P19 protein (OD_600_ = 0.6) and *A. tumefaciens* cells harboring a vector expressing the rice stripe virus (RSV)-encoded NSvc4 protein (OD_600_ = 0.6). For other treatments, *A. tumefaciens* harboring the vector expressing RSV NSvc4 was replaced by *A. tumefaciens* harboring the vector expressing the new virus NS, N, or NSs protein, or *A. tumefaciens* harboring the vector expressing β-glucuronidase (GUS). The mixed *A. tumefaciens* cells were infiltrated into *N. benthamiana* leaves. At 6 dpi, the infiltrated leaves were examined under a UV light or a confocal microscope.

### 2.8. Western Blot Analysis

The total protein of *N. benthamiana* leaves was extracted in a protein extraction buffer [30], followed by Western blot analysis as previously described [31]. In brief, the resulting protein samples were separated on 12.5% sodium dodecyl sulfate–polyacrylamide gel electrophoresis (SDS–PAGE). After protein bands were transferred to nitrocellulose membranes, the membranes were blocked for 1 h in a 0.01 M phosphate-buffered saline (PBS) containing 5% skimmed milk at room temperature, followed by a 60 min incubation in a primary antibody solution at 37 °C. After three rinses in PBS containing 0.05% Tween-20, the membranes were incubated for 60 min in the horseradish peroxidase (HRP)-labeled secondary antibody solution at 37 °C. The detection signal was visualized by adding the Immobilon Western Chemiluminescent HRP Substrate, according to the manufacturer’s instructions (Millipore, WBKLS0100).

### 2.9. Transmission Electron Microscopy

Field-collected rice plants were analyzed for viral infections using RT-PCR, after which leaf tissues were sampled from the diseased rice plants and fixed for 4 h in a 2.5% glutaraldehyde solution (pH 7.0) at room temperature. The fixed leaf samples were dehydrated in serially diluted ethanol solutions. After 20 min of incubation in acetone, the tissues were individually embedded in the Spurr resin (SPI Supplies, Westchester, PA, USA). Ultra-thin sections were cut from the embedded samples and examined under a Hitachi H-7650 transmission electron microscope.

### 2.10. Confocal Microscope

The agro-infiltrated leaves were cut from the *N. benthamiana* plants and observed under a confocal microscope (model FLUOVIEW FV3000; Olympus, Tokyo, Japan). The excitation wavelengths of GFP, RFP, and aniline blue were 488, 555, and 405 nm, respectively, and the emission wavelength was set at 510 nm for GFP, 583 nm for RFP, and 461 nm for aniline blue.

## 3. Results

### 3.1. Identification of a Novel Rice Dwarf-Associated Bunya-like Virus

During the surveys of rice viral diseases in 2019, rice plants showing dwarfing, reduced tillering, and blighted grain symptoms were found in paddy fields in the Zhejiang province of China, and samples were obtained (Figure 1A). Using the total RNA from these field-collected rice plants as the template and primers designed according to the known rice-infecting viruses in China, common rice viruses, including rice dwarf virus (RDV), rice black-streaked dwarf virus (RBSDV), southern rice black-streaked dwarf virus (SRBSDV), rice gall dwarf virus (RGDV), rice stripe mosaic virus (RSMV), rice ragged stunt virus (RRSV), and rice stripe virus (RSV) were separately monitored by RT-PCR. RT-PCR results found no known rice virus in the sampled rice plants (Appendix A). Therefore, we conducted further analyses using RNA-seq and RT-PCR technologies. Analyses of the resulting sequencing reads with the TopHat software v2.1.1 produced three long contigs with 1093, 1681, and 6483 nucleotides (nt), respectively. BLASTx searching the GenBank database with these three long contigs identified a new virus belonging to the order *Bunyavirales*. These three long contigs corresponded to the large (L), medium (M), and small (S) segments of bunyaviruses, respectively. To confirm this RNA-seq result, we amplified partial segments of the above three long contigs using RT-PCR, and all gene segments could be amplified (Figure 1B–D). We then performed 5′ and 3′ RACEs to obtain the full-length viral genome. The 5′ and 3′ RACEs and sequencing results showed that these three full-length viral genomic RNAs contained 6562 (L), 1667 (M), and 1120 (S) nt, respectively. Complete genome sequences of RDaBV have been deposited in the GenBank under the accession numbers ON366507 (L), ON366508 (M), and ON366509 (S). Based on the typical viral symptoms and its genome, we tentatively named this new virus rice dwarf-associated bunya-like virus (RDaBV). In the 2021 rice-growing season, we also detected RDaBV and known rice viruses (RBSDV, SRBSDV, RDV, RRSV, RGDV, RSMV, and RSV) in rice plants in paddy fields in the Zhejiang province of China. We found that 4 of 30 field-collected rice plants showing dwarfing symptoms were only infected with RDaBV, with an infection rate of RDaBV reaching 13.3% (Appendix A). This indicates that RDaBV is already one of the prevalent rice viruses in Zhejiang province, China.

An analysis of leaf tissues from the RDaBV-infected rice plants through transmission electron microscopy (TEM) showed that these leaf samples contained enveloped spherical virus-like particles approximately 125 nm in diameter (Figure 1E), similar to the morphology of some known bunyaviruses. The diameters of most bunyavirus virions range from 80 to 120 nm, but some bunyaviruses have larger virions. For instance, the High Plains wheat mosaic virus in the family *Fimoviridae* has virions up to 200 nm in diameter [32] and the Hantaan virus in the family *Hantaviridae* has virions 80–200 nm in diameter [33].

### 3.2. Viral Genome and Protein Prediction

Similar to other plant-infecting bunyaviruses, the RDaBV genome was found to have three RNA segments (Figure 2A). The RDaBV L segment shares 21.96–51.24% nucleotide homology with viruses in the family *Discoviridae* (Table 1). The L segment contains a large ORF (nucleotide position 85 to 6483) that is predicted to encode a protein with 2132 amino acids (aa). Analysis of this protein using the Conserved Domain Database (CDD) tool in the NCBI showed that this protein contains a conserved Bunya_RdRp super family domain (cl20265, aa residue 684 to 1238). Therefore, we considered that the RDaBV L segment also encodes an RNA-dependent RNA polymerase (RdRp). The RdRp protein encoded by RDaBV shares 27.69–57.39% aa identity with that encoded by the viruses in the family *Discoviridae* (Table 1). The BLASTx search using the RDaBV L segment query indicated that this RdRp shares the highest aa sequence similarity (57.86%) with that of the Botrytis cinerea negative-stranded RNA virus 2 (GenBank accession QJT73695.1) (Appendix A).

The RDaBV M segment shares 33.23–47.83% nucleotide homology with viruses in the family *Discoviridae* (Table 1). The M segment of RDaBV contains 1667 nt with a single ORF that is predicted to encode an NS protein with 506 aa residues. The NS protein shares the aa sequence identity ranging from 10.67% to 35.48% with that of viruses in the family *Discoviridae*. The BLASTx search using this M segment query showed that all matched sequences are from viruses in the family *Discoviridae* (Appendix A). The CDD search result showed that the RDaBV NS protein had no conserved protein domain.

The RDaBV S segment shares 37.7–40.76% nucleotide homology with viruses in the family *Discoviridae* (Table 1). As shown in Figure 2, the RDaBV S segment is ambisense and contains two ORFs, one on the viral RNA strand and another on the viral RNA complementary strand. This phenomenon also occurs in some other bunyaviruses. For example, S segments of viruses in both *Phenuiviridae* and *Tospoviridae* families are ambisense, encoding for a small nonstructural protein (NSs) and a nucleocapsid (N) protein [34]. The ORF1 on the viral RNA strand of the RDaBV S segment (nt position 95–442) was predicted to encode the NSs protein with 115 aa residues, and the ORF2 on the viral RNA complementary strand (nt position 174–1031) was predicted to encode the N protein with 285 aa residues (Figure 2A). The aa sequence identity of the N protein between RDaBV and the known viruses in *Discoviridae* ranged from 30.53 to 50.17% and was much higher than that between RDaBV and viruses of other families in the order *Bunyavirales* (Table 1). At the same time, the matched sequences by the BLASTp search using the RDaBV N protein query were all from the family *Discoviridae* (Appendix A). The RDaBV NSs protein did not match any viral proteins in the NCBI protein database by the BLASTp search.

Moreover, the sequence alignment result of the RDaBV three genomic segments found that both the 3′ and 5′ ends of all three RDaBV genomic segments have a complementary conserved sequence CACACAAAG, and the same 14 and 16 nucleotides at their 3′ and 5′ ends, respectively (Figure 2B). This characteristic is shared by some plant-infectious bunyaviruses, such as melon chlorotic spot virus (MeCSV) and rice grassy stunt virus (RGSV) [35,36].

### 3.3. Phylogenetic Analysis of RDaBV

To determine the evolutionary relationship of RDaBV, we retrieved 52 RdRp aa sequences of viruses from 14 families in the order *Bunyavirales* and one family (*Rhabdoviridae*) in the order *Mononegavirales* in the NCBI database (Appendix A). We then constructed a phylogenetic tree using aa sequences of the downloaded 52 RdRps and RDaBV RdRp (Figure 3). The phylogenetic tree indicated that the RDaBV is in the same branch as viruses from the family *Discoviridae* (Figure 3). RDaBV was evolutionarily closer to the Plasmopara viticola lesion-associated mycobunyavirales-like virus 8. This result is consistent with the viral protein identity analyses (Table 1). The family *Discoviridae* was created by the ICTV in July 2021. The hosts of all six reported viruses in the new family *Discoviridae* are from the phylums *Oomycota* and *Ascomycota*. However, the host of RDaBV is rice.

### 3.4. Pathogenicity and Subcellular Localization of Viral Proteins

To clarify the symptom determinant of RDaBV, we constructed three PVX-based recombinant vectors: PVX-N, PVX-NS, and PVX-NSs. These three constructed vectors were individually agro-infiltrated into *N. benthamiana* leaves; *A. tumefaciens* carrying PVX was used as the negative control. After 7 days post-agro-infiltration (dpi), we found that PVX-N and PVX-NS caused plant leaf necrosis symptoms, while PVX-NSs showed the same mosaic symptoms as PVX (Figure 4A). We further detected the infection of PVX through Western blot using the PVX CP specific antibody, and our results demonstrated that the accumulation levels of PVX in *N. benthamiana* leaves infiltrated with PVX-NSs or PVX-N were lower than that in leaves infiltrated with PVX, and the accumulation levels of PVX in *N. benthamiana* leaves infiltrated with PVX-NS were higher than that infiltrated with PVX (Figure 4B). Furthermore, the accumulation level of PVX in the leaves infiltrated with PVX-N was significantly reduced (Figure 4B). This suggests that proteins N and NS may be symptom determinants of RDaBV.

To determine subcellular localization patterns in plant cells of viral proteins, RDaBV *NS*, *N*, and *NSs* genes were separately subcloned into the transient expression vector pCambia 1300-GFP to obtain pCambia 1300-NS-GFP (NS:GFP), pCambia 1300-N-GFP (N:GFP), and pCambia 1300-NSs-GFP (NSs:GFP). *N. benthamiana* leaves were individually agro-infiltrated with these expression vectors, and the green fluorescence was observed at 48 h post-agro-infiltration (hpi). The green fluorescence (from the NSs:GFP, NS:GFP, and N:GFP fusion proteins) was located at the cytomembranes, cytoplasm, and nuclei of *N. benthamiana* leaf cells, which is similar to that of GFP on its own (Figure 5). Furthermore, NS:GFP and N:GFP fusion proteins formed puncta on the cytomembrane and in the cytoplasm (Figure 5).

### 3.5. RDaBV NSs Protein Functions as the Viral Movement Protein

To determine which viral protein encoded by RDaBV can function as a viral movement protein in the plant, we produced three expression vectors that can express FLAG-tagged NSs (NSs-flag), FLAG-tagged N (N-flag), and FLAG-tagged NS (NS-flag). *A. tumefaciens* cells carrying the NSs-flag, NC-flag, NS-flag, RSV NSvc4 (the positive control), or the GUS expression vector (the negative control) were mixed with *A. tumefaciens* cells carrying the PVX-GFPΔp25 expression vector and *A. tumefaciens* cells carrying the TBSV P19 expression vector. The mixed *A. tumefaciens* cells were individually infiltrated into *N. benthamiana* leaves. At 5–7 dpi, the infiltrated leaves were harvested and examined under a confocal microscope and a UV lamp. The result showed that in the presence of NSs-flag, the mutant PVX-GFPΔp25 moved into the neighboring cells, similar to that observed in leaves co-expressing PVX-GFPΔp25 and RSV NSvc4 (Figure 6A,B). This indicates that NSs can function as a viral movement protein (MP). However, both N and NS proteins could not complement the cell-to-cell movement of mutant PVX-GFPΔp25 in infiltrated *N. benthamiana* leaves (Figure 6A). We collected the infiltrated *N. benthamiana* leaves and analyzed them with Western blot using a GFP-specific antibody to further confirm this finding. Our results demonstrated that in the presence of NSs-flag, more GFP had accumulated in the leaf tissues, which is similar to that in leaves co-expressing PVX-GFPΔp25 and RSV NSvc4 (Figure 6C). The Western blot result with the anti-flag antibody showed the expressions of GUS-flag, NSs-flag, N-flag, and NS-flag fusion proteins (Figure 6C). However, because the expressed NSvc4 did not fuse a flag tag, NSvc4 could not be monitored by the anti-flag antibody (Figure 6C).

Previous studies have demonstrated that plant virus infections can induce callose depositions around plasmodesmata (PD) to inhibit virus cell-to-cell spread, while viral MPs can reduce callose depositions near PDs [37,38]. To further investigate the function of RDaBV NSs, we transiently expressed *NSs-flag* or *GUS* in *N. benthamiana* leaves. Two days later, the infiltrated leaves were harvested, stained with aniline blue, and examined under a confocal microscope. We found that leaves expressing *NSs* had fewer aniline blue-stained dots than those expressing *GUS* (Figure 7A). We counted the number of fluorescence dots in individual cells and confirmed that the leaves expressing *NSs* had fewer fluorescence dots compared to the leaves expressing *GUS* (Figure 7B). To investigate whether RDaBV NSs can target PD in cell walls, we co-expressed *NSs-GFP* and TMV *MP-RFP* fusion proteins in *N. benthamiana* leaves through agro-infiltration. Two days later, the leaves were treated with a 1 M sucrose solution followed by a confocal microscopy observation. As the TMV MP (red punctate signal) shows the position of PDs in cell walls, our results provide support for the hypothesis that RDaBV NSs can target PDs (Figure 7C, white arrows).

### 3.6. RDaBV NS Protein Roles as an RNA Silencing Suppressor

*A. tumefaciens* harboring a vector expressing GFP was mixed with *A. tumefaciens* harboring a vector expressing RDaBV *NS-flag*, *N-flag*, *NSs-flag*, *P19*, or *GUS*. The mixed *A. tumefaciens* combinations were then individually infiltrated into transgenic 16c *N. benthamiana* leaves. In this experiment, *A. tumefaciens* harboring the vector expressing *GUS* and *P19* was used as the negative and positive control. At 6 dpi, the leaves co-expressing *GFP* and *GUS*, *GFP* and *N-flag*, *GFP* and *NSs-flag*, *GFP* and *NS-flag*, or *GFP* and *P19* were harvested and photographed under a UV light. We found that *GFP* silencing was suppressed in the presence of RDaBV NS or P19 (Figure 8A). On the other hand, *GFP* silencing occurred in the presence of RDaBV N, NSs, or GUS (Figure 6A). The Western blot result showed that leaves co-expressing *GFP* and *NS-flag* or *GFP* and *P19* accumulated more GFP compared to leaves co-expressing *GFP* and *NSs-flag*, *GFP* and *N-flag*, or *GFP* and *GUS* (Figure 8B), indicating that RDaBV NS is a viral suppressor of RNA silencing.

## 4. Discussion

It is well known that the number of known plant-infecting viruses accounts for a small fraction of the total number of plant-infecting viruses in nature. In our field survey in 2019, rice plants showing dwarfing, reduced tillering, and blighted grain symptoms, which are similar to that shown by virus-infected rice plants, were found in several rice fields in Zhejiang province, China. Because analyses of the collected rice samples for known virus infections through RT-PCR failed, we speculated that these rice samples could be infected with unidentified virus(es). To test this speculation, we analyzed these rice samples using RNA-seq. The RNA-seq and RT-PCR results showed that these dwarf rice plants contained an ambisense-segmented viral RNA genome, similar to that reported for viruses in the order *Bunyavirales*. Based on the phylogenetic relationship, we tentatively named this new virus the rice dwarf-associated bunya-like virus (RDaBV).

The organization of the RDaBV genome shares similar characteristics with that of viruses in the family *Discoviridae* in the order *Bunyavirales*. The RDaBV genome consists of three RNA segments: L, M, and S. The genome organization of RDaBV was determined according to the existence of conserved motifs and previously reported features of bunyaviruses [1,2,3,4]. The L segment encodes an RdRp, and the M segment encodes a nonstructural protein (NS). The RDaBV S segment encodes a nucleocapsid and another nonstructural protein (NSs). The termini of three RDaBV RNA segments have the same complementary conserved sequence CACACAAAG, a feature shared by other bunyaviruses, including melon chlorotic spot virus and rice grassy stunt virus [35,36].

Though some plant viruses are diffused by contact and seeds [39], the majority of plant viruses are transmitted by vectors, including insects, mites, nematodes, or fungi [40]. Of these vector-transmitted plant viruses, most are transmitted via insect vectors in fields [41]. In this study, we tried to transmit RDaBV to rice plants using insect vectors, including the small brown planthopper (*Laodelphax striatellus*), brown planthopper (*Nilaparvata lugens*), white-backed planthopper (*Sogatella furcifera*), and *Recilia dorsalis*, but all failed (data not shown). A phylogenetic analysis using RDaBV RdRp showed that this novel rice virus belonged to the newly created family *Discoviridae* and was evolutionarily close to Plasmopara viticola lesions-associated mycobunyavirales-like virus 8 [25]. The hosts of all six reported viruses in the family *Discoviridae* are from the phylum *Oomycota* or *Ascomycota*. Given that we did not find the insect vector of RDaBV, we suspect that the novel virus could be transmitted by fungi but could also be transmitted by other insects.

The encoding of MPs is one of the most important characteristics of plant viruses. The plasmodesmata (PD) is the critical macromolecule transport corridor in plant cell walls. Because of its size exclusion limit (SEL), PD is a bottleneck during plant virus infections [42]. Many studies have demonstrated that plant virus infections can induce callose accumulation around PDs, leading to a restriction of virus intercellular trafficking. On the other hand, plant virus-encoded MPs have been shown to interact with ankyrin repeat-containing proteins (ANK) to reduce callose accumulation, relax callose sphincters, and increase the SEL of PDs [37,38]. In this study, we identified the NSs protein as the MP of RDaBV by the movement complementation and callose staining experiment, which demonstrates that RDaBV is a plant-infected virus. At the same time, during the survey of rice viral diseases in 2021, we collected 30 rice plants showing dwarfing symptoms in paddy fields in the Zhejiang province of China and monitored RDaBV and the known rice viruses in these samples by RT-PCR. We found that four rice plants showing dwarfing symptoms were only infected with RDaBV, and the infection rate of RDaBV reached 13.3% (Appendix A).

We also found that the NS protein can act as a viral RNA silencing suppressor. RNA silencing (RNAi) is a common defense strategy used by plants to resist viral invasions. This anti-viral strategy can be interfered with by viral suppressors of RNA silencing (VSRs) encoded by plant viruses via inhibiting dsRNA recognition, interfering with RNA-induced silencing complex (RISC) assembly, directly binding to proteins crucial to the RNA silencing pathway, and interfering with silent signal transmission [43,44,45,46]. However, in this study, we do not know which step of plant RNA silencing was suppressed by the RDaBV NS protein.

In summary, we identified a novel rice-infecting bunya-like virus, RDaBV, belonging to the family *Discoviridae* of the order *Bunyavirales* in Zhejiang province, China. We determined that the RDaBV NSs protein is a viral movement protein and its NS protein is a viral RNA silencing suppressor. Since vectors play important roles in rice virus spread in fields, further studies are needed to determine how RDaBV is transmitted in rice fields. Furthermore, the pathogenicity of RDaBV must be verified using Koch’s postulate. Additionally, its evolutionary history and impact on rice production must be further studied. Nonetheless, the findings presented in this study enrich our understanding of bunyaviruses and rice viruses and can contribute to more effective strategies for controlling this rice viral disease.

## Figures and Tables

**Figure 1 viruses-14-01183-f001:**
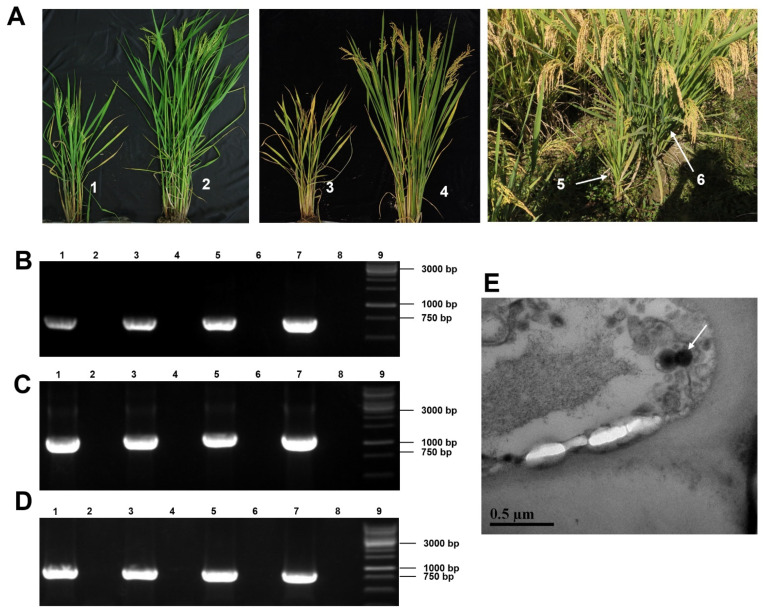
Rice plant symptoms, virions, and RT-PCR detection of the rice dwarf-associated bunya-like virus (RDaBV). (**A**) Field-collected rice plants showed dwarfing symptoms. Nos. 1, 3, and 5 indicate dwarf plants, and Nos. 2, 4, and 6 indicate non-infected plants. (**B**–**D**) RT-PCR detected RDaBV in rice plants using specific primer pairs for RDaBV genomic L, M, and S segments. Lanes 1–6 were field-collected rice plants, as shown in Figure 1A. Lane 7 was an RDaBV-infected rice plant used as a positive control. Lane 8 was a healthy rice plant used as a negative control. Lane 9 was a 1 kb DNA marker. (**E**) Suspected virus particles of RDaBV were observed using transmission electron microscopy. Scale bar = 5 µm.

**Figure 2 viruses-14-01183-f002:**
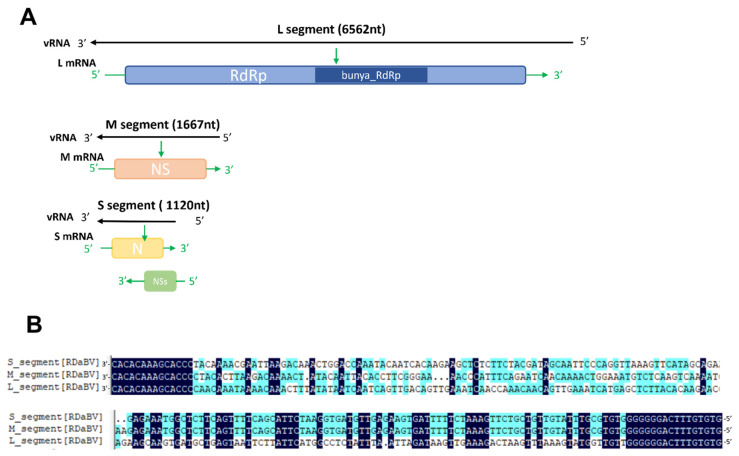
The genomic characteristics of RDaBV. (**A**) Schematic diagram of RDaBV genome organization. (**B**) Alignment of 3′ and 5′ ends of three genome strands of RDaBV.

**Figure 3 viruses-14-01183-f003:**
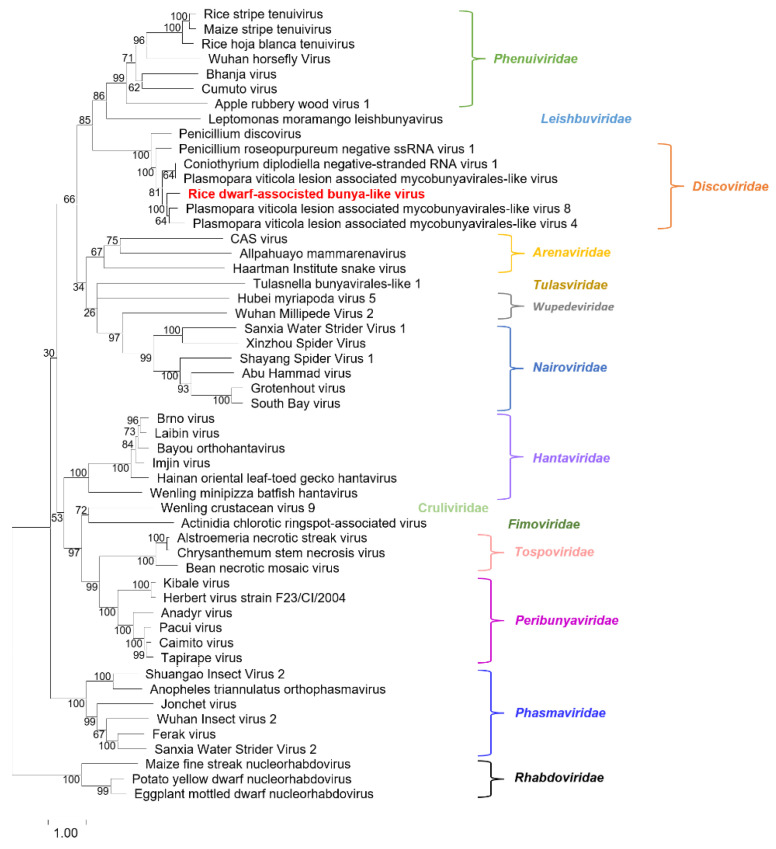
Phylogenetic tree analysis of the RNA-dependent RNA-polymerase of RDaBV and representative viruses from the orders *Bunyavirales* and *Mononegavirales*. The tree was inferred using the maximum likelihood method and a JTT matrix-based model with 1000 bootstraps. The number represents the bootstrap value of each branch. Viruses from the same family are shown in the same color. The name RDaBV is shown in red. Detailed information about the viruses used in the analysis is listed in Appendix A.

**Figure 4 viruses-14-01183-f004:**
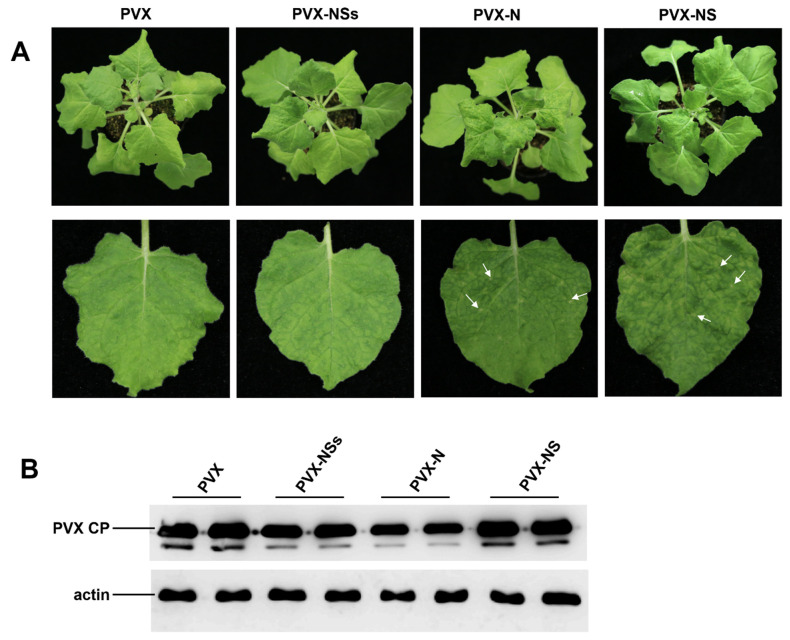
Symptom determinant analysis of RDaBV. (**A**) Heterologous expression of RDaBV NSs, N, and NS proteins. *N. benthamiana* leaves were infiltrated with *A. tumefaciens* harboring PVX-NSs, PVX-N, PVX-NS, or PVX construction. White arrows indicate necrotic patches. (**B**) Western blot analyzing PVX accumulation levels in *N. benthamiana* leaves. Actin was used as an internal reference.

**Figure 5 viruses-14-01183-f005:**
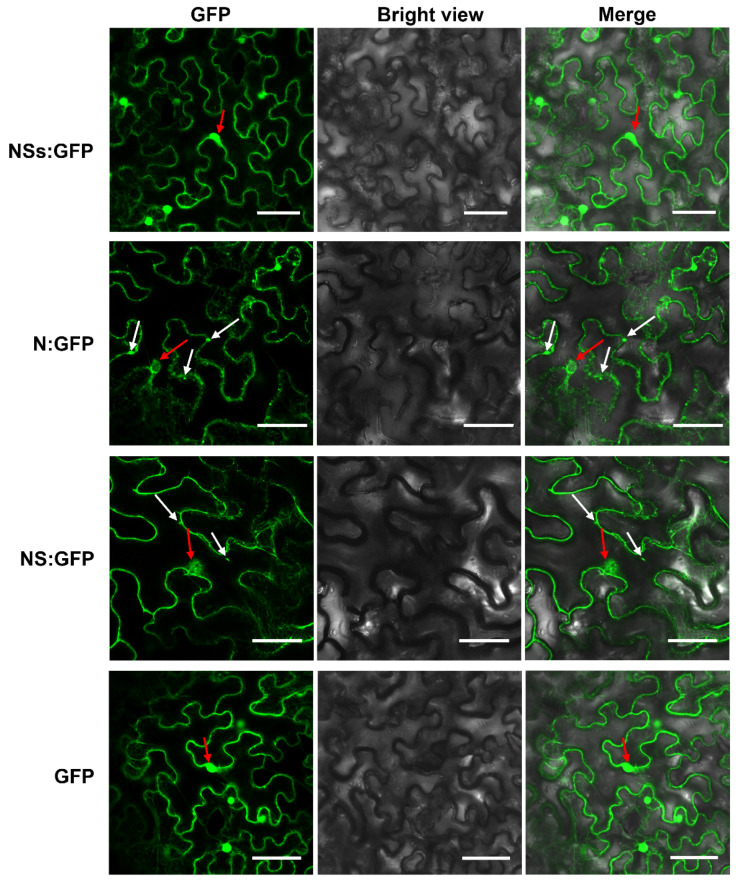
Subcellular localization analyses of RDaBV NSs, N, and NS proteins. *N. benthamiana* leaves were infiltrated with *A. tumefaciens* EHA105 harboring NSs-GFP, N-GFP, NS-GFP, or GFP construction. Scale bars, 50 µm. White arrows indicate GFP aggregates, while red arrows indicate nuclei.

**Figure 6 viruses-14-01183-f006:**
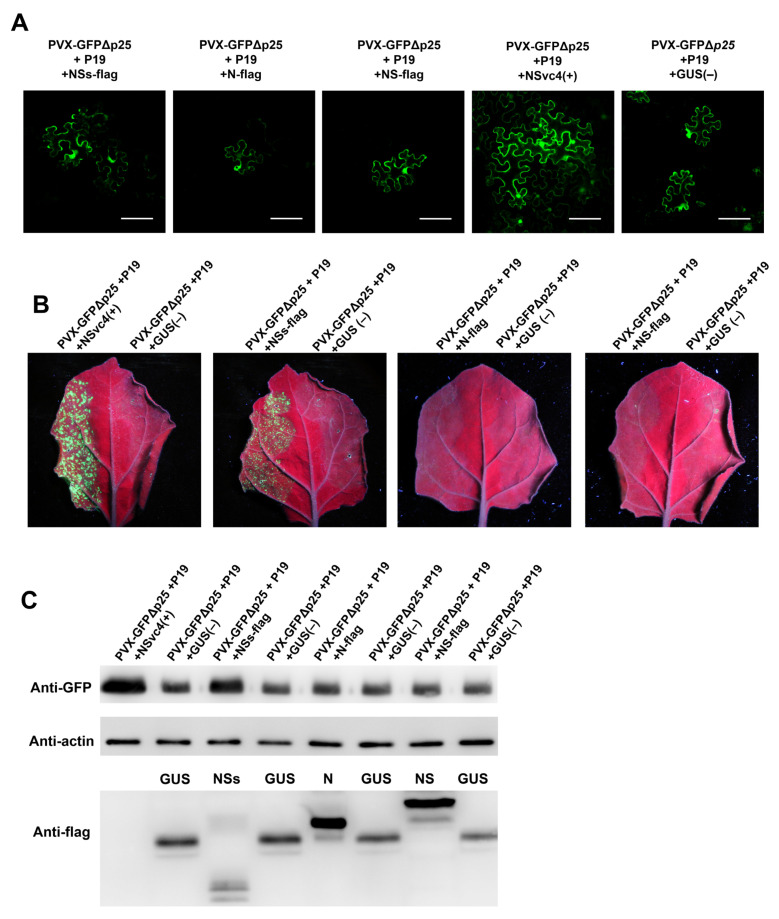
RDaBV NSs complement the intercellular movement of PVX-ΔP25. (**A**) RDaBV NSs could complement the intercellular movement of PVX-ΔP25. The co-infiltrated leaves were observed under a confocal microscope at 6 dpi. Bars, 100 μm. (**B**) GFP fluorescence of co-infiltrated *N. benthamiana* leaves was observed by a UV lamp at 6 dpi. (**C**) Western blot analyzing GFP and viral protein expression levels in co-infiltrated *N. benthamiana* leaves. Actin was used as the internal reference.

**Figure 7 viruses-14-01183-f007:**
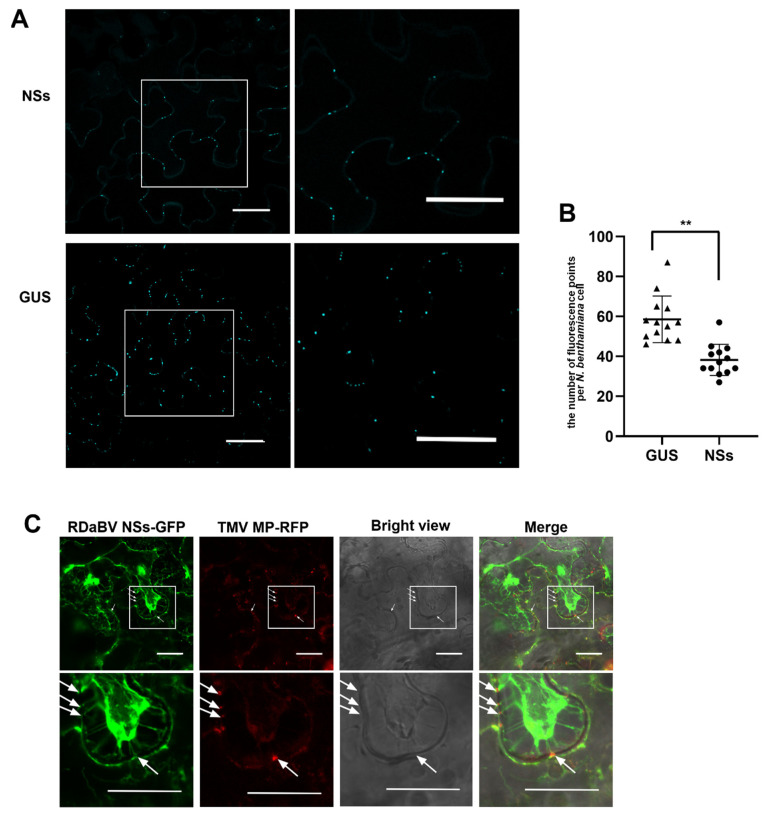
RDaBV NSs can reduce the callose deposition and locate plasmodesmata. (**A**) Aniline blue staining of callose of the leaves expressing RDaBV NSs. The right column shows the enlarged images of the boxed areas in the images in the left column. Bars, 100 µm. (**B**) Statistics of fluorescence puncta in a single cell of *N. benthamiana* leaves expressing *NSs* and *GUS*. The triangles and dots indicate the number of puncta in a single cell of *N. benthamiana* leaves expressing *GUS* and *NSs*, respectively. Error bars showed the SD for 13 individual cells; ** *p* < 0.01 by *t*-test analysis. (**C**) Colocalization of RDaBV NSs-GFP and TMV MP-RFP in co-infiltrated *N. benthamiana* leaves was observed using laser confocal microscopy. The leaves were treated with 1 M sucrose solution at 2 dpi. White arrows indicate the plasmodesmata positions. The lower row is the enlarged image of the boxed area in the upper image. Bars, 20 μm.

**Figure 8 viruses-14-01183-f008:**
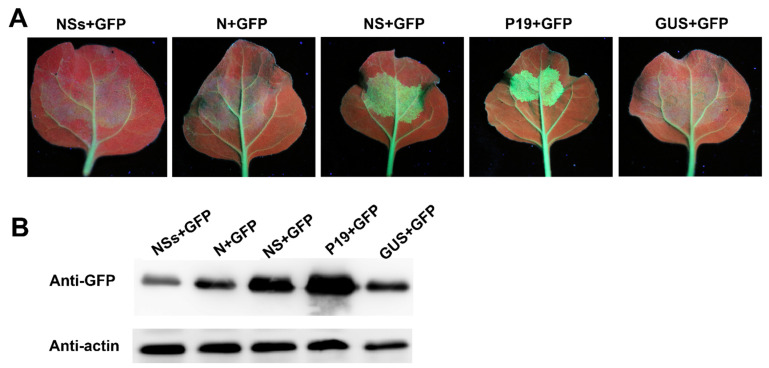
NS protein encoded by RDaBV is a post-transcriptional gene silencing suppressor. (**A**) GFP fluorescence of co-infiltrated *N. benthamiana* leaves was observed with a UV lamp at 6 dpi. (**B**) Western blot analyzing GFP protein expression levels in co-infiltrated *N. benthamiana* leaves. Actin was used as the internal reference.

**Table 1 viruses-14-01183-t001:** The nucleotide and amino acid identities between RDaBV and the representative species in 14 families of the order *Bunyavirales*.

Virus Name	Order	Family	Nucleotide Identities (%)			Amino Acids Identities (%)			
			L Segment	M Segment	S Segment	RdRp	NS Protein	N Protein	NSs Protein
Allpahuayo mammarenavirus	*Bunyavirales*	*Arenaviridae*	39.7	None *	14.21	9.61	7.84	4.99	8.7
Wenling crustacean virus 9	*Bunyavirales*	*Cruliviridae*	41.01	21.58	28.33	8.89	5.34	7.72	None
Actinidia chlorotic ringspot-associated virus	*Bunyavirales*	*Fimoviridae*	39.71	30.3	28.04	9.51	8.41	7.37	7.08
Penicillium roseopurpureum negative ssRNA virus 1	*Bunyavirales*	*Discoviridae*	38.96	39.64	38.02	45.36	18.27	33.24	None
Penicillium discovirus	*Bunyavirales*	*Discoviridae*	51.24	33.23	39.87	39.92	10.67	30.53	None
Coniothyrium diplodiella negative-stranded RNA virus 1	*Bunyavirales*	*Discoviridae*	39.02	41.01	38.35	49.32	23.91	37.23	None
Plasmopara viticola lesion associated mycobunyavirales-like virus 9	*Bunyavirales*	*Discoviridae*	39.03	42.11	37.7	49.32	25.09	48.26	None
Plasmopara viticola lesion associated mycobunyavirales-like virus 8	*Bunyavirales*	*Discoviridae*	39.96	47.83	40.76	57.39	35.48	42.24	None
Plasmopara viticola lesion associated mycobunyavirales-like virus 4	*Bunyavirales*	*Discoviridae*	21.96	46.15	38.98	27.69	30.25	50.17	None
Laibin virus	*Bunyavirales*	*Hantaviridae*	41.22	18.26	25.04	10.66	5.24	5.85	None
Leptomonas moramango leishbunyavirus	*Bunyavirales*	*Leishbuviridae*	38.83	29.86	25.27	9.07	8.64	6.64	None
Hubei myriapoda virus 5	*Bunyavirales*	*Mypoviridae*	28.7	20.97	17.81	6.74	6.06	7.55	None
Grotenhout virus	*Bunyavirales*	*Nairoviridae*	18.59	None	12.71	4.22	None	5.63	None
Abu Hammad virus	*Bunyavirales*	*Nairoviridae*	22.89	14.74	24.96	4.95	4.12	8.2	None
South Bay virus	*Bunyavirales*	*Nairoviridae*	20.35	None	8.04	4.89	None	5.12	None
Shayang Spider Virus 1	*Bunyavirales*	*Nairoviridae*	21.82	14.85	23.19	4.99	4.46	6.36	None
Sanxia Water Strider Virus 1	*Bunyavirales*	*Nairoviridae*	23.25	17.43	20.75	4.88	4.56	5.97	None
Xinzhou Spider Virus	*Bunyavirales*	*Nairoviridae*	23.02	21.79	23.23	5.24	4.4	6.32	None
Kibale virus	*Bunyavirales*	*Peribunyaviridae*	38.32	27.02	38.9	9.28	5.06	8.42	None
Ferak virus	*Bunyavirales*	*Phasmaviridae*	39.57	16.35	30.21	8.79	3.33	6.54	6.96
Bhanja virus	*Bunyavirales*	*Phenuiviridae*	40.24	20.76	25.33	9.73	4.3	6.67	3.51
Cumuto virus	*Bunyavirales*	*Phenuiviridae*	40.73	22.06	37.45	9.7	5.27	8.07	None
Wuhan horsefly Virus	*Bunyavirales*	*Phenuiviridae*	30.21	24.76	24.58	7.63	6.23	9.47	4.72
Apple rubbery wood virus 1	*Bunyavirales*	*Phenuiviridae*	39.38	40.38	35.29	8.23	6.92	13.2	None
Rice hoja blanca virus	*Bunyavirales*	*Phenuiviridae*	31.36	18.99	21.19	8.21	5.16	7.52	6.5
Rice stripe tenuivirus	*Bunyavirales*	*Phenuiviridae*	32.26	20.33	18.08	8.66	5.63	11.49	8.15
Alstroemeria necrotic streak virus	*Bunyavirales*	*Tospoviridae*	31.65	14.55	15.16	7.01	4.85	10.14	6.59
Tulasnella bunyavirales-like 1	*Bunyavirales*	*Tulasviridae*	22.52	None	None	8.65	5.22	7.02	None
Wuhan Millipede Virus 2	*Bunyavirales*	*Wupedeviridae*	24.94	10.12	23.27	5.68	2.6	6.25	None

* None, no submitted sequence.

## Data Availability

Not applicable.

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
