# Peer review of "Discovery and Genomic Function of a Novel Rice Dwarf-Associated Bunya-like Virus"

_viruses, 2022, doi:10.3390/v14061183_

Round 1

Reviewer 1 Report

Wang et al. found a new Bunya-like virus in rice when conducting an epidemiological search in the paddy in Zhejiang Province, China. The finding of a new symptom-induced virus lays a foundation for controlling virus diseases in the field and improving rice production. Generally, the manuscript is organized nicely and easy to follow, though grammar mistakes frequently show up and some figures' quality is low. Below, please find the comments that will require the authors' attention.
Major concern:
1, In Figure 2A, the length of the three fragments should be aligned. For example, the scheme of the L RNA stands for 6562. Therefore, the M RNA and S RNA scheme length should be around 1/4 and 1/5 of that of the L RNA. Given that the authors marked the nt number of the L RNA in the scheme, this note should be added to M and S RNA. In Figure 2B, please mark the 3' and 5' ends.

2, Paragraphs 3.3 and 3.4 share the same title. So, undoubtedly, the title for 3.4 is a typo.

3, In Figure 3, the family name of each virus showed up twice in the phylogenetic tree. Please delete the one following the virus name. In addition, please indicate the meaning of the number in the figure legend.

4, In Figure 4B, please denote which band represents the PVX-CP since there are two bands in the upper layer. In addition, the biological significance of comparing the expression of PVX in paragraph 3.4 is not clear. If this experiment aims to confirm the expression of target proteins, please give the western blot data. From my point of view, the necrosis phenotype is insufficient to clarify that N and NS are the symptom determinant of RDaBV. The RDaBV-infected rice plants exhibit dwarfing, reduced tillering, and blighted grain symptoms, different from necrosis. More explanation is needed.

5, The quality of Figure 5 is relatively low. It is easy to tell the cytomembranes and cytoplasm with the GFP signal. However, the N and NS trigger granules in the cells, so it is hard to distinguish the nuclei and the "puncta." Please mark these organelles in a whole cell.

6, The quality of Figure 6A is still low. It is not easy to tell if there is a foci in the first panel. In addition, in figure 6B, the authors should provide data for other infiltration combinations presented in figure 6A. If the authors want to confirm the expression of the Flag-tagged proteins, please provide the western blot data.

7, In Figure 7C, TMV MP is used as a marker for plasmodesmata. However, as far as I know, TMV MP could localize in cell wall/plasma membrane-rich fractions and branched plasmodesmata. Thus, TMV MP might not be a good marker here. I suggest the author directly stains the cells expressing NSs with Aniline blue to confirm this claim that NSs localized in PD, though the NSs somehow reduced the accumulation of callus in the cells.

8, In Figure 8B, the expression of NSs seems to reduce the expression of GFP compared to that of the GUS control. In contrast, in Figure 8A, the GFP signal is more robust in the image of NSs+GFP compared to that of the GUS+GFP.

Minor concerns:

1, Line 20 and 22. Please provide the full name for N protein and NS protein in the abstract. 

2, Line 40, Reference 4, "Zhang, W.; Wu, T.: Guo, M" should be deleted.

3, Latin name and plasmid name should be Italic. I suggest the author go through the whole manuscript. I only list a few.
 Line 49, Panaeus chinensis; Line 63, Oryza sativa; Line 169, 178, 359, 403, A. tumefaciens; Line 220, 269 Bunyavirales; Line 241, Fimoviridae; Line 263, Botrytis cinerea; Order and Family names in Table 1;  Line 340, pCambia 1300.

4, Line 307, "Plasmopara viticola lesion associated with mycobunyavirales-like virus 8", delete "with".

5, Line 51, Reference/ PLos Pathogen to "PLoS Patho." Line 567, delete "Advanced online publication"

Author Response

Wang et al. found a new Bunya-like virus in rice when conducting an epidemiological search in the paddy in Zhejiang Province, China. The finding of a new symptom-induced virus lays a foundation for controlling virus diseases in the field and improving rice production. Generally, the manuscript is organized nicely and easy to follow, though grammar mistakes frequently show up and some figures' quality is low. Below, please find the comments that will require the authors' attention.

OUR RESPONSE: We would like to thank the respected reviewer 1 for your positive comments and valuable suggestions. We have carefully revised my manuscript according to your comments and suggestions.
Major concern:
1, In Figure 2A, the length of the three fragments should be aligned. For example, the scheme of the L RNA stands for 6562. Therefore, the M RNA and S RNA scheme length should be around 1/4 and 1/5 of that of the L RNA. Given that the authors marked the nt number of the L RNA in the scheme, this note should be added to M and S RNA. In Figure 2B, please mark the 3' and 5' ends.

OUR RESPONSE: We would like to thank the respected reviewer 1 for your correct suggestions. In this revised version, We have modified the Figure 2A and 2B according to your suggestions.

2, Paragraphs 3.3 and 3.4 share the same title. So, undoubtedly, the title for 3.4 is a typo.

OUR RESPONSE: The wrong title of 3.4 has been corrected.

3, In Figure 3, the family name of each virus showed up twice in the phylogenetic tree. Please delete the one following the virus name. In addition, please indicate the meaning of the number in the figure legend.

OUR RESPONSE: In this revised version, we have deleted the family name following the virus name and the number represents the bootstrap value of each branch.

4, In Figure 4B, please denote which band represents the PVX-CP since there are two bands in the upper layer. In addition, the biological significance of comparing the expression of PVX in paragraph 3.4 is not clear. If this experiment aims to confirm the expression of target proteins, please give the western blot data. From my point of view, the necrosis phenotype is insufficient to clarify that N and NS are the symptom determinant of RDaBV. The RDaBV-infected rice plants exhibit dwarfing, reduced tillering, and blighted grain symptoms, different from necrosis. More explanation is needed.

OUR RESPONSE: In Figure 4B, the PVX-CP band has been denoted. Because we have no antibodies against viral proteins and the expressed viral proteins have no tag, we cannot provide the Western blot result to confirm the expression of the three viral proteins of RDaBV. But, according to the symptoms of inoculated plants, we think that the viral proteins were expressed by PVX. We agree with your viewpoint that the necrosis phenotype is insufficient to clarify that N and NS are the symptom determinants of RDaBV. So, in this version, we have changed it to the N and NS may be symptom determinants of RDaBV.

5, The quality of Figure 5 is relatively low. It is easy to tell the cytomembranes and cytoplasm with the GFP signal. However, the N and NS trigger granules in the cells, so it is hard to distinguish the nuclei and the "puncta." Please mark these organelles in a whole cell.

OUR RESPONSE: In revised version, we have improved Figure 5 and marked the nuclei with red arrows.

6, The quality of Figure 6A is still low. It is not easy to tell if there is a foci in the first panel. In addition, in figure 6B, the authors should provide data for other infiltration combinations presented in figure 6A. If the authors want to confirm the expression of the Flag-tagged proteins, please provide the western blot data.

OUR RESPONSE: In revised version, we have improved Figure 6A according to your comments and suggestions. At the same time, we have provided photographs for other infiltration combinations in Figure 6B and added the Western blot result of the expressed viral proteins of RDaBV in Figure 6C.

7, In Figure 7C, TMV MP is used as a marker for plasmodesmata. However, as far as I know, TMV MP could localize in cell wall/plasma membrane-rich fractions and branched plasmodesmata. Thus, TMV MP might not be a good marker here. I suggest the author directly stains the cells expressing NSs with Aniline blue to confirm this claim that NSs localized in PD, though the NSs somehow reduced the accumulation of callus in the cells.

OUR RESPONSE: In 1987, Tomenius et al first reported tobacco mosaic virus (TMV) movement protein (MP) was localized to the plasmodesmata of infected tobacco leaves by immunogold cytochemistry [1]. Now, TMV MP is one of the best-characterized viral MPs, which strongly targets to plasmodesmata in newly infected cells and in transgenic plants [2]. Therefore, TMV MP is widely used as the marker for plasmodesmata [3-5]. Thus, we think TMV MP is enough to prove the localization of PD.

[1] Tomenius, K.; Clapham, D.; Meshi, T. Localization by immunogold cytochemistry of the virus-coded 30K protein in plasmodesmata of leaves infected with tobacco mosaic virus. Virology. 1987, 160(2), 363–371.

[2] Lucas WJ. Plant viral movement proteins: agents for cell-to-cell trafficking of viral genomes. Virology. 2006 Jan 5;344(1):169-84.

[3] Rodrigues, K. B.; Orílio, A. F.; Blawid, R.; Melo, F. L.; Nagata, T. Subcellular localization of p29, a putative movement protein of pepper ringspot virus. Arch Virol. 2015, 160(1), 359–364.

[4] Thomas, C. L.; Bayer, E. M.; Ritzenthaler, C.; Fernandez-Calvino, L.; Maule, A. J. Specific targeting of a plasmodesmal protein affecting cell-to-cell communication. PLoS Biol. 2008, 6(1), e7.

[5] Jiang, J.; Kuo, Y. W.; Salem, N.; Erickson, A.; Falk, B. W. Carrot mottle virus ORF4 movement protein targets plasmodesmata by interacting with the host cell SUMOylation system. New Phytol. 2021, 231(1), 382–398.

8, In Figure 8B, the expression of NSs seems to reduce the expression of GFP compared to that of the GUS control. In contrast, in Figure 8A, the GFP signal is more robust in the image of NSs+GFP compared to that of the GUS+GFP.

OUR RESPONSE: in Figure 8A, the GFP signal is more robust in the image of NSs+GFP compared to that of the GUS+GFP, which is caused by the different exposure time of photographs. In revised version, we have improved Figure 6A.

Minor concerns:

1, Line 20 and 22. Please provide the full name for N protein and NS protein in the abstract. 

OUR RESPONSE: Revised accordingly.

2, Line 40, Reference 4, "Zhang, W.; Wu, T.: Guo, M" should be deleted.

OUR RESPONSE: Revised accordingly.

3, Latin name and plasmid name should be Italic. I suggest the author go through the whole manuscript. I only list a few.
 Line 49, Panaeus chinensis; Line 63, Oryza sativa; Line 169, 178, 359, 403, A. tumefaciens; Line 220, 269 Bunyavirales; Line 241, Fimoviridae; Line 263, Botrytis cinerea; Order and Family names in Table 1; Line 340, pCambia 1300.

OUR RESPONSE: Revised accordingly. Line 263, Botrytis cinerea negative-stranded RNA virus 2 is a virus name, so it's not italicized. As far as we know, plasmid names in papers from this journal are not italicized, thus, we don't italicize plasmid names.

4, Line 307, "Plasmopara viticola lesion associated with mycobunyavirales-like virus 8", delete "with".

OUR RESPONSE: Revised accordingly.

5, Line 51, Reference/ PLos Pathogen to "PLoS Patho." Line 567, delete "Advanced online publication"

OUR RESPONSE: Revised accordingly.

Reviewer 2 Report

In this paper, high throughput RNA-seq was used to identify a bunya-like virus in rice plants showing the dwarfing symptom. This virus was tentatively named rice dwarf-associated Bunya-like virus (RDaBV). The authors revealed genome organization that is typical for plant bunya-like viruses. RNA-L encodes RdRp, showing closest similarity to fungi virus Botrytis cinerea negative-stranded RNA virus 2, RNA-M codes for single NS protein, whereas RNA-S is ambisense and encodes NSs and N proteins. The Potato Virus X-bases expression system with inserted N, NS and NSs ORFs showed that N protein and NS protein are potential symptom determinants of RDaBV, whereas movement complementation results indicated that NSs is a viral movement protein (the latter suggestion was confirmed by callose staining experiment).

In general this paper is logically good and well written. I suggest to accept this manuscript as in its present form.

Author Response

In this paper, high throughput RNA-seq was used to identify a bunya-like virus in rice plants showing the dwarfing symptom. This virus was tentatively named rice dwarf-associated Bunya-like virus (RDaBV). The authors revealed genome organization that is typical for plant bunya-like viruses. RNA-L encodes RdRp, showing closest similarity to fungi virus Botrytis cinerea negative-stranded RNA virus 2, RNA-M codes for single NS protein, whereas RNA-S is ambisense and encodes NSs and N proteins. The Potato Virus X-bases expression system with inserted N, NS and NSs ORFs showed that N protein and NS protein are potential symptom determinants of RDaBV, whereas movement complementation results indicated that NSs is a viral movement protein (the latter suggestion was confirmed by callose staining experiment). In general, this paper is logically good and well written. I suggest to accept this manuscript as in its present form.

OUR RESPONSE: We would like to thank the respected reviewer 2 for your positive comments.